# Mental Health of Young People in the Post-Pandemic Era: Perspective Based on Positive Psychology and Resilience

**DOI:** 10.3390/ijerph22101574

**Published:** 2025-10-15

**Authors:** Daniel T. L. Shek

**Affiliations:** Department of Applied Social Sciences, The Hong Kong Polytechnic University, Hong Kong SAR 999077, China; daniel.shek@polyu.edu.hk

**Keywords:** adolescents, youths, mental health, post-pandemic era, PYD, family resilience, interdisciplinary

## Abstract

With the gradual decline in COVID-19 cases, there is a need to re-visit the mental health of adolescents and emerging adults in the post-pandemic period. Several observations can be highlighted from the scientific literature. First, while some studies suggest that mental health of young people has worsened in the post-pandemic period, there are inconsistent and conflicting findings. Second, there are more studies on psychological morbidity than on positive psychological attributes. Third, compared with the West, there are relatively fewer Chinese studies. Fourth, compared with adolescents, there are relatively fewer studies on emerging adults. Based on these observations of the existing literature, I have detailed several reflections on the mental health of young people, including enhancing positive psychological attributes in young people through positive youth development (PYD) programs, building up the individual resilience of young people, strengthening family resilience, adopting multidisciplinary, interdisciplinary and transdisciplinary approaches in understanding the mental health of young people, building more well-articulated theoretical models, charting future research directions, and developing intervention strategies in the post-pandemic period.

## 1. Mental Health of Young People in the Post-Pandemic Era

There are two views on the impact of the COVID-19 pandemic on the mental health of young people in the post-pandemic era. The first perspective argues that as COVID-19 is stressful and young people have to make many adjustments in response to the pandemic, the mental health of young people would be negatively affected, and there will be mental health problems in the post-pandemic era. This view is consistent with the notion of “post-traumatic stress” that trauma created by COVID-19 would create stress that has negative impacts on the mental health of young people. The second view holds that the pandemic constitutes a growth opportunity for young people through which they can further develop their resilience, therefore resulting in better mental health. This is consistent with the existential and humanistic views that life adversities and challenges are facilitators for personal growth and positive development.

With reference to mental health problems, there are studies supporting the first view that mental health problems of young people are worsening in the pandemic and post-pandemic periods [1,2,3,4,5,6,7,8,9,10,11,12]. Kapadia [13] discussed the public health consequences of COVID-19 in early 2024. He pointed out that based on different sources of evidence (e.g., Centers for Disease Control and Prevention), roughly three-quarters of adolescents in the United States showed poor mental health during the pandemic. He further argued that social distancing, coupled with geopolitical and economic challenges, amplified the mental health problems of young people. Using data collected from university students in three countries, Bersia et al. [14] showed that roughly two-thirds of the respondents in Italy and Spain displayed excessive psychological symptoms, and roughly half of the respondents in all countries showed high depression scores. Close et al. [15] also outlined the stressors and psychological issues in the post-pandemic era; Ning et al. [16] showed that sleep problems, anxiety and stress were challenges faced by university students in the post-pandemic period. As commented by Su et al. [17], COVID-19 had a “detrimental effect on the lives of a very large proportion of the young people” (p. 1).

One stressor that contributed to the poor mental health of young people during the post-pandemic period is school resumption and academic adjustment problems. Young and Young [18] concluded that the academic performance of the students was behind the pre-pandemic levels and there was a slow recovery. Based on Malaysian students, Asadullah [19] reported that while many students had positive emotions of going back to school and there were reductions in negative emotions such as worries, there were signs of negative learning, such as struggling to learn and loss in learning, and most students reported that they did not receive support. Schwartz et al. [20] reported that students expressed moderate concern for health, family and family confinement when they went back to school, and roughly one-fourth of the students showed excessive stress. In addition, Banerjee et al. [21] suggested school re-opening was met by school reluctance that was triggered by anxiety, hostility and secondary gain. In short, the negative impact of COVID-19 into the post-pandemic era is captured by Hampton-Anderson [22] that “over the past decade, our global community has faced a growing number of shocks. As these challenges persist into the future, their impact on the mental health and well-being of adolescents cannot be ignored.” (p. 159).

On the other hand, there are research findings supporting the second view in the scientific literature. For example, Chen et al. [23] showed that there was a decrease in pandemic panic, anxiety and depression (but an increase in PTSD and suicidal thoughts) in the post-pandemic period. Based on four waves of data, Novaković et al. [24] showed that probable adjustment disorder and depression were highest in 2020/2021, and there was a “slight improvement” across time. However, they also pointed out that global crises may continue to negatively impact adolescent mental health in the post-pandemic era. Li et al. [25] showed that psychological morbidity indexed by the General Health Questionnaire decreased in the post-pandemic period. In another longitudinal study, Ruetter et al. [26] showed that while there was an initial increase in psychological strain in the first pandemic peak, there was post-pandemic “improvement” in mental health later on. The researchers explained this increase in terms of high pre-pandemic psychological challenges faced by the respondents and their coping during the pandemic. In addition, it is possible that the pandemic may help to nurture the resilience of young people, which may eventually help to reduce their psychological morbidity.

Regarding positive mental health indicators, there are comparatively fewer studies that also produce conflicting findings [27]. For example, there are studies showing that there was a reduction in life satisfaction over the pandemic period [28]. However, Nakagawa et al. [29] showed that life satisfaction of the adolescent and adult respondents was stable across the COVID-19 period. Schriek et al. [30] also showed that there was an initial rise and then a subsequent drop in emotional exhaustion in the pandemic and post-pandemic period, while enthusiasm remained stable.

There are several explanations for the conflicting findings on the mental health of adolescents and emerging adults in the post-pandemic period. First, as COVID-19 is stress-inducing, it is reasonable to see that the mental health of young people deteriorated in the initial period, and then became relatively better with the gradual decline of the lethality of COVID-19. Second, different assessment tools, samples, research designs and timing of the studies may contribute to the observed findings. Third, as young people may actually learn new skills and coping strategies under the pandemic [31], this may contribute to the mental health of young people. Hence, there is a need to further understand the reasons behind the “gradual improvement” phenomenon. Finally, despite the fact that there had been a prolonged increase in mental health problems in young people before the pandemic, failure to control for pre-pandemic mental health status may contribute to the inconsistencies observed in the related studies.

## 2. Observations Based on the Related Studies Captured by PsycInfo

To understand the research on the mental health of young people in the post-pandemic era, I conducted a literature search on mental health studies based on adolescents and emerging adults in the post-pandemic era using PsycInfo (see Table 1 and Table 2). There are several observations based on the data collected that echo those reported previously [27]. First, for studies on mental health problems, studies mainly concentrated on depression and anxiety. Second, compared with studies on mental health problems, there are comparatively fewer studies on positive mental health. Third, relatively fewer studies have been conducted in the Chinese context, except in the area of Internet addiction. Fourth, there are comparatively more studies on high school students than university students and emerging adults.

There are several implications of the findings in Table 1 and Table 2. First, there is a need to step up research on positive mental health. One possible factor contributing to this phenomenon is that there are more self-report measures of mental health problems and existing measures of positive mental health commonly do not have “cutoff” scores. Hence, it would be helpful to develop more validated measures of positive mental health, which can give more precise information with reference to the concept of ‘at-risk’ mental health. Another contributing factor to this observation is that mental health is still primarily conceived in terms of psychological morbidity. Second, as most studies concentrate on the Western context, researchers have to bear in mind that Western studies may be “WEIRD” studies, which typically involve samples recruited from Western, educated, industrial, rich and democratic societies. Data collected from non-Western contexts, particularly in China, is important for three reasons. First, China has a huge population that is important as far as the generalizability of research findings is concerned. Second, there are cultural variations in the manifestations of mental health symptoms. For example, depressive symptoms in Chinese people commonly take the form of somatization because Chinese people are more reserved in expressing their emotions. Third, there are indigenous Chinese concepts of mental health, such as emphasis on interpersonal harmony and fulfillment of duties as components of gratification. The third observation is that studies on Internet addiction were mainly conducted in Chinese subjects. This suggests that Internet addiction may be a growing problem in China, particularly during the pandemic and post-pandemic periods. As the COVID-19 pandemic has forced schools to lockdown and students mainly communicated with others through the Internet in the first two years of the pandemic, it is necessary to further explore the impact of the pandemic on Internet addiction. Finally, as there are more studies on high school students than university students and emerging adults, the findings suggest that researchers need to know more about the mental health of emerging adults who may be more negatively affected by COVID-19 in the non-academic domains such as unemployment and economic constraints. There are different challenges faced by high school adolescents and emerging adults studying in colleges. First, while high school students search for their identity, university students consolidate their identity. Second, high school students are more dependent on their parents than are emergent adults, who are more capable of making independent decisions. Third, it is commonly expected that university students face more occupational challenges than do high school students. As such, more related research based on emerging adults should be conducted.

Although there are variations in the related prevalence findings in the post-pandemic era, the disturbing mental health problems lead to two reflections. The first reflection concerns the factors that may help to buffer the negative impact of the pandemic on the developmental outcomes of young people. With reference to this reflection, researchers may look at how positive psychological attributes may help to moderate the negative impact of the pandemic on the well-being of young people. Second, there is a need to re-think prevention strategies, particularly the enhancement of individual resilience and family resilience. This is particularly important in view of rising global crises, such as global warming, geopolitical conflict, economic instability and health challenges. The human race experienced the Spanish flu roughly one century ago. We then experienced SARS in 2003 and COVID-19 in 2019. Without a crystal ball, we can still certainly say that pandemics will appear again in the future. Then, how to prepare the society, especially young people, to thrive under a future pandemic is important. These issues are discussed in the following reflections.

### 2.1. Reflection 1: Positive Psychological Attributes in the Post-Pandemic Period

The first reflection is on the “protective factors” for mental health during the pandemic and post-pandemic periods, particularly the role of positive psychological attributes [27]. There are studies showing that positive psychological attributes play an important role in reducing the negative impact of the pandemic stress on the psychological well-being of young people [32,33]. Researchers also argue for stepping up positive psychological attributes which can help to promote individual resilience during the COVID-19. For example, Waters et al. [34] suggested that several positive psychological attributes are particularly important during the pandemic to buffer the negative impact of the pandemic on mental health outcomes. These included life purpose, coping style, self-compassion, courage, gratitude, character strengths, positive affect and interpersonal relationships, and quality connection.

Slavich et al. [35] outlined the key positive psychological attributes promoting resilience and personal growth during the pandemic, which included nurturance of social belongingness, compassion, and kindness. For social belonging, it refers to one’s self-others connection and social group affiliation. Belongingness is important during the pandemic because social isolation and social distancing in the pandemic decreased social belongingness and one’s sense of community. Regarding compassion, it refers to one’s identification with the suffering of other people and self (i.e., compassion towards others and self-compassion). In fact, compassion towards others can be regarded as a form of “covert” helping behavior that one is trying to help, such as through prayers and blessing others. For self-compassion, one learns to accept one’s inadequacies, limitations and imperfections under the pandemic. Finally, kindness is voluntary prosocial acts such as serving as volunteers, helping needy people and donating money. This act of giving help is to fulfill one’s responsibility under the pandemic and it promotes social capital, which would eventually contribute to community and individual mental health.

Tonis [36] used a randomized controlled trial to test the impact of a positive psychology self-help intervention on the well-being of participants exhibiting anxiety and depression problems. Based on data collected at three time points, findings showed that the intervention program had significant positive effects on measures of adaptation and different measures of mental health, spiritual well-being, and self-compassion at different time points.

In the post-pandemic era, it is helpful to consider how positive psychological attributes may enhance mental health and protect individuals from the adverse effects of stress arising from the pandemic, which would build up the inner strengths of young people to face future pandemics.

### 2.2. Reflection 2: Individual Resilience as a Primary Prevention Strategy

The second reflection is on prevention, particularly on the promotion of individual resilience in young people so that they can be better prepared for future pandemics [27]. While different ecological factors, such as community cohesion, may facilitate the development of individual resilience, personal factors also play an important role in building up individual resilience. There are studies showing that individual resilience was positively related to mental health amongst young people in the post-pandemic era [37].

Building individual resilience is an important primary prevention strategy. Through building up individual resilience, individuals can have more inner strengths and psychosocial competence to cope with adversity arising from global crises. This strategy is similar to the common prevention approach of encouraging the public to have regular exercise to avoid chronic illnesses such as a heart attack. There are two main approaches to enhancing individual resilience. The first approach is to develop and implement specific resilience training programs. For example, Pinto et al. [38] reviewed children and adolescent resilience programs, including 17 studies for qualitative analyses and 13 studies for quantitative analyses. They concluded that the programs under review were generally effective in enhancing resilience and the benefits were maintained up to six months after program termination.

Another approach to enhance adolescent resilience is to strengthen the general developmental assets in young people based on the positive youth development literature [39,40,41], which would eventually enhance the resilience of young people. In the developmental assets model proposed by Peter Benson [42,43], there are 20 external assets and 20 internal assets that can enhance the holistic development of young people, which further consolidates young people’s ability to thrive. For external assets, there are four domains. The first one is support (support from family, healthy family communication, positive relationship with other adults, caring neighborhood, caring school ecology, and parental involvement in schooling). The second one is youth empowerment, where young people perceive that they are valued by others and they are resources, serving others and providing safety. The third domain is boundaries and expectations, which includes boundaries in family, school, and neighborhood, role models of adults, positive peer influence and high expectations. The final domain is constructive use of time, which includes creative activities, youth programs, religious community, and time at home. There are also four domains for the internal assets. The first domain is commitment to learning, which includes motivation to achieve, engagement with school, homework, school bonding and interest in reading. The next domain is positive values, which includes sense of care, equality and social justice, integrity, honesty, responsibility and restraint. The third domain refers to social competencies, which include skills to plan and make decisions, social skills, cultural competence, resistance skills and skills to resolve conflict in a peaceful manner. The final domain is positive identity, which includes recognition of personal power, self-esteem, purpose in life, and positive views of the future.

In addition, building developmental assets is also emphasized in the 5C model of Richard Lerner [44,45]. These include connection (e.g., good family relationship), competence (e.g., development of psychosocial skills), confidence (e.g., sense of control and ability to attain goals), character (e.g., integrity and kindness) and caring (e.g., engagement in voluntary work and helping others). Regarding competences as developmental assets, the social-emotional learning (SEL) model proposed by CASEL similarly highlights the importance of competence, including self-awareness, social awareness, self-management, responsible decision making and social relationships [46,47,48,49].

Finally, based on a review of PYD programs in the United States, Richard Catalano and his associates [50] found that besides resilience, there are several other PYD constructs that contribute to positive development in young people. These include psychosocial skills (cognitive, social, behavioral, emotional and moral competencies), self-determination, self-efficacy, spirituality, positive identity, belief in the future, prosocial norms, prosocial involvement, bonding, and recognition for positive behavior. There are studies showing that PYD programs are able to promote school adjustment and reduce negative behavior. With particular reference to the Chinese context, PYD programs exemplified by the P.A.T.H.S. program were found to enhance holistic youth development, including their resilience [51,52].

### 2.3. Reflection 3: Family Resilience as a Primary Prevention Strategy

While individual resilience is important, it is noteworthy that individual resilience is shaped by factors outside an individual [27]. For example, Panzeri et al. [53] reported that while facilitators of individual resilience included social distancing and psychological factors (e.g., trait resilience, conscientiousness), factors impairing resilience included COVID-19 symptoms (anxiety, PTSD), intolerance of uncertainty, loneliness, living with children, and living in regions where the virus was starting to spread.

Based on ecological models, besides individual resilience, it is also important to build up family resilience [27]. There are studies showing that the pandemic negatively impacted family processes and dynamics and family moderated the negative impact of pandemic stress on well-being [54,55,56,57,58,59,60,61,62]. Hence, the third reflection is on how to promote family resilience, which is a protective factor for individual mental health. Based on family systems theory, family stress model and family resilience concepts, Prime et al. [63] proposed a model outlining the processes that influence the well-being of caregivers, holistic family functioning, and child developmental outcomes during the pandemic. The model posits that COVID-19 creates social disruption, which eventually impairs child adjustment through caregiver well-being. It is argued that individual, dyadic and family systems should be taken into account to understand the impact of the COVID-19 pandemic on child mental health, where family resilience within the family structure system moderates the direct and indirect effects of the pandemic on child developmental outcomes. Finally, pre-existing family vulnerabilities or pre-existing strengths would exacerbate or buffer the above-mentioned processes. In short, family resilience plays a pivotal role in the influence of the COVID-19 pandemic on child developmental outcomes.

Walsh [64] proposed that there are three domains in family resilience, including family belief systems, family organizational patterns and family communication and problem solving. Within each domain, there are three family processes. For family belief system, it forms the “cognitive structure” for making sense of adversity and possible coping. There are three processes under the family belief system. The first one is forming meaning about adversity (e.g., positive appraisal, meaning making, attributions and expectations). Family belief is important because it shapes the family climate as well as the coping behavior of the family under adversity. The second family process is a positive outlook, which includes hope to overcome the crisis, strength-based coping, getting the most out of the worst, and focusing on the bright side of adversity. The final process is on transcendence and spirituality, focusing on values and purpose, faith, connections with others and nature, growth and positive transformation under adversity, inspirations and aspirations.

The second domain is on the organizational processes within the family. The first process is flexibility, which refers to the adaptation to change, re-organization of the family to respond to challenges, and flexible family forms through cooperation and mutual respect. The second process is connectedness, which includes mutual respect, support, cooperation, commitment as well as reconciliation. The final process is mobilization of social and economic resources. These include family, social and community support, setting good examples, maintaining financial security, and transactions with the larger social systems, such as receiving institutional support.

The final domain of family resilience is concerned about communication and problem-solving processes. The first process is clarity, which includes clear, consistent and predictable messages and swift clarification of confused and unclear messages. The next family process is openness in emotional sharing, including sharing painful feelings such as sadness, anger, anxiety, guilt and disappointment, and positive affect (e.g., love, appreciation, gratitude and kindness). The final family process is collaborative problem-solving, which includes co-creation of solutions, collective decision making, negotiation and goal setting, learn from success and failure experiences, and preparing for the future.

There are studies showing that family resilience plays an important role in shaping the mental health of young people during the pandemic. Based on 2691 Chinese adolescents, Zhuo, Yu and Shi [65] examined the influence of family resilience on adolescent mental health, with perceived pandemic stress as a mediator and meta mood as a moderator of the prediction of family resilience on perceived pandemic stress. While family resilience positively predicted adolescent mental health, perceived pandemic stress negatively predicted adolescent mental health. In another study examining individual and family resilience predictors in mental health (depression, anxiety and stress), Chan, Piehler and Ho [66] recruited 442 and 597 respondents in the United States and Hong Kong, respectively, in the early stage of the pandemic. They found that both individual resilience and family resilience were positively linked to positive mental health. In Minnesota, individual resilience moderated the negative impact of pandemic-related stressors on depressive symptoms. At the same time, family communication and problem-solving also moderated the link between pandemic-related stressors and stress symptoms. However, while family resilience measures were negatively related to mental health problems measured in participants in Hong Kong, families with a higher family positive outlook unexpectedly showed a stronger relationship between pandemic-related stress and anxiety symptoms.

Ho et al. [67] recruited 200 family dyads to examine the role of family resilience on COVID-19 threat perception, impact and exposure. They found that the psychological impact of a family member was influenced by dyadic financial impact, and family resilience was negatively associated with the psychological impact of COVID-19. Based on the findings, they argued for the importance of family intervention and an ecological understanding of family risk and protective factors under the pandemic. Based on the responses of Indonesian families during the COVID-19 pandemic, Ramadhana [68] reported findings on the emotional reactions and family resilience. They found that there were significant relationships amongst the different measures of family resilience. Different family resilience measures were also negatively related to anger and anxiety.

In a review of family resilience studies during the pandemic period, Gayatri and Irawaty [69] highlighted the importance of family resilience, particularly daily practices of gratitude, good and healthy communication, and positive family activities in building family cohesion, trust and happiness, which would eventually help families to thrive under the pandemic. Obviously, such practice would be helpful in the post-pandemic period. Shek, Leung and Tan [27,70] also argued that policies on the promotion of mental health in the post-pandemic era should step up family resilience on top of individual resilience.

### 2.4. Reflection 4: Multidisciplinary, Interdisciplinary and Transdisciplinary Approaches to Mental Health

The fourth reflection regards the use of different disciplinary knowledge to understand and improve adolescent mental health. Primarily, mental health can be understood in terms of a single discipline. For example, while neuroscience researchers attempt to understand which parts of the brain are related to adolescent depression, psychologists examine learning processes underlying adolescent mental health issues, and sociologists are interested in examining what social factors (e.g., poverty and social class) are related to mental health problems. Obviously, an understanding of adolescent mental health issues from a single discipline is not enough. Hence, researchers in different disciplines may adopt cross-disciplinary or multi-disciplinary approaches to join together to work on a mental health problem. The arguments for understanding human behavior and social phenomena through an interdisciplinary approach are presented by many researchers in the scientific literature [71,72,73,74,75,76,77,78,79,80]. Moitra et al. [81] highlighted the importance of adopting an ecological understanding in looking at system changes such as family resilience and systemic changes.

For cross-disciplinary research, researchers commonly borrow concepts, theories and/or research methods to solve a problem. The mode typically involves unidirectional influence where researchers in a single field “cross-over” to another field, such as applying statistical methods to solve psychology problems, or using sociological concepts (e.g., social capital) to understand psychological events (e.g., addictive behavior). In short, it involves the transfer of ideas and/or tools across fields. For multi-disciplinary research, while multiple disciplines attempt to solve a shared problem, researchers work independently without much interaction and integration. It is an “additive model” by simply adding findings from different disciplines. For example, in an attempt to understand adolescent substance abuse, psychologists focus on cognitive distortions, educators focus on school climate, and sociologists focus on inequalities as antecedents of adolescent substance abuse. In essence, this is an independent contribution from different disciplines without blending.

In contrast, an interdisciplinary approach focuses on integration of concepts, theories, research methods, findings, interventions, insights and/or policies. For example, public health professionals blend drug research, psychological counseling, family therapy, community support and policy advocacy to develop a multi-level integrative intervention model to help adolescents with depression. Another example is to involve pediatric, social work, occupational therapy, family, policy, and education researchers to devise social policies to help poor adolescents. In short, interdisciplinary research involves integration of theories, methods and findings from different disciplines to produce innovative frameworks (e.g., integration and combination of genetic and behavioral data) with deep integration to generate new insights and full synthesis of the findings, while respecting and maintaining the boundaries of disciplines involved. For example, Costello et al. [82] reported an interdisciplinary study to protect young people from the harmful effects of social media. Lygre et al. [83] also proposed an interdisciplinary intervention program with the involvement of different health professionals in intervention programs for children and adolescents, which may lead to lower fragmented care, medical errors and sub-optimal health outcomes.

If we take one step further, we can involve not just academic and research colleagues but also non-academic stakeholders, which is more transformative and holistic in nature. This is commonly referred to as a transdisciplinary approach, which transcends academic boundaries with the involvement of non-academic stakeholders. It commonly focuses on the impact in real life and co-creation of knowledge by different stakeholders aiming at the development of transformative societal solutions. In essence, transdisciplinary research is beyond academia, which is different from the interdisciplinary approach in terms of attributes and goals. Stakeholder involvement, focus on problems in the real world, and generation of new knowledge and solutions are unique to the transdisciplinary approach, where scientists, corporates, Government officials and the local community are involved to create solutions to help poor kids. For example, Ramaswamy et al. [84] proposed a transdisciplinary SAMVAD (support, advocacy and mental health interventions for children in vulnerable circumstances and distress) model for child and adolescent mental health. Basically, SAMVAD has four elements, including the development of standardized child and adolescent mental health (CAMH) protection protocols, strengthening CAMH knowledge and skills, implementation of research, and development of scalable CAMH models.

### 2.5. Reflection 5: Theoretical Considerations

There are several unfinished tasks with reference to theories, research and intervention in the mental health of young people. Primarily, researchers have to sharpen theories on the mental health of young people with reference to different systems [27]. In particular, there is a need to look at the mental health of young people from a more systemic and holistic perspective [85]. For example, regarding school reluctance, Banerjee et al. [21] suggested that intervention in the school system, such as familiarization with the school environment, building positive emotions, and cultivation of positive school experience alone may be helpful but not enough. They pointed out that “previous research has linked school closures to reductions in learning. However, policymakers should also consider children’s social and emotional needs. Governments should be sensitive to the current challenges for children and provide flexibility in terms of pedagogy, assessments, and curriculum for a limited time to support them to cope” (p. 1). In short, it is beneficial to adopt a holistic perspective by taking different systems into account. In addition, as mental health exists in a cultural context, it would be helpful to develop indigenous theories. Second, researchers have to re-think about the relationship between the protective factors and the mental health of young people. For example, while it is commonly asserted that there is a linear relationship between family functioning and mental health, too low or too high levels of family cohesion or flexibility may impair the well-being of family members.

### 2.6. Future Research Directions

Regarding research on the mental health of young people in the post-pandemic era, there are several possible directions. First, it is important to conduct more research on the mental health of young people using multidisciplinary, interdisciplinary and transdisciplinary research. This research orientation is particularly important as the mental health of young people is multi-faceted and there are different levels of understanding. Second, there is a need to re-visit and clarify the inconsistent findings on the mental health with particular reference to the potential moderators such as research designs, time of data collection and outcome indicators. For example, Seaborn et al. [86] highlighted several issues in the studies on psychological resilience during COVID-19. Third, more mental health studies on emerging adults in non-WERID countries should be conducted. Fourth, more research on how geopolitics influences the mental health of young people should be conducted. For example, Alnaser et al. [87] reported that the Gaza conflict impaired the psychological health of the Kuwait population, who shared similar cultural and geopolitical factors with those in the conflict zone. Finally, Yiu et al. [88] identified three issues surrounding post-pandemic research and policy, including novel versus traditional techniques, centralized versus local decision making, and possibilities of collaboration amongst academic disciplines and other stakeholders.

### 2.7. Future Intervention Directions

It is always better to prevent than to cure. While intervention aiming at treating the mental health problems of young people arising from the pandemic (such as PTSD) is important, it is equally important to build up the individual resilience of young people so that they can endure and thrive under global crises such as the pandemic [27]. Regarding prevention, while prevention was mainly physical (e.g., wearing masks) and financial (e.g., unemployment subsidies) during the pandemic, comparatively less emphasis was placed on psychosocial prevention. As mentioned above, the cultivation of individual resilience can protect individuals. In fact, there are programs in the pandemic, such as positive youth development programs, that could help to promote resilience in young people. In addition, enhancement of family resilience is also an important prevention strategy because it is regarded as an antecedent of individual resilience as well as adolescent mental health issues [89].

Based on the experience of the pandemic, there are many suggestions for mental health services and policies. For example, Shidhaye [90] argued that mental health problems are prevalent in adolescence, and adolescents did not receive adequate evidence-based interventions even before the pandemic and the problem intensified during the pandemic. They highlighted the importance of access to mental health services and suggested that school-based interventions and digital health technologies, with specific reference to equity and identifying those who need the service most, would be helpful. In addition, they also suggested using a common theory, standardized tools and methodology, and affordable evidence-based intervention. Chandra et al. [91] proposed a mental health systems framework to understand the content of mental health systems research to identify the gaps in COVID-19 research, and suggest a direction for future research paradigm. There are also studies focusing on the use of digital technologies and telework [92,93].

Stepanova et al. [94] reviewed the mental health services responding to COVID-19 and identified several changes. These include the adoption of telecommunication technologies, which have created much impact, such as access and delivery improvements, influence of technological literacy on provider-patient relationship, increase in care inequality such as the disconnection in marginalized groups, and use of telehealth in mental health care as a promising tool. Husain et al. [95] also highlighted the transition from face-to-face psychiatric service to virtual mental health consultation (i.e., telepsychiatry). Similarly, Palinkas [96] examined the impact of the COVID-19 pandemic on mental health policy and service. They proposed that there should be more multi-level collaboration and building formal and informal networks, integration of face-to-face and telehealth, flexible reimbursement, licensing and supervision arrangement, and regular data collection leading to data-driven decision making.

Looking into the future, there is a need to re-think mental health services and policy from a transdisciplinary lens. Taylor et al. [97] highlighted several principles in prioritizing research on youth mental health. The first principle is the adoption of an assets-based approach in which young people are regarded as equal partners rather than burdens of the society or passive service recipients. The second principle is to shift from a “deficit” approach to one that focuses on the capabilities of people that can be utilized at different levels in the service delivery model. The third principle is to focus on reciprocal and mutual responsibilities between service providers and clients. The fourth principle is to promote peer support networks that can complement professional input, particularly in the area of knowledge transfer. The fifth principle is to minimize the distinction between professionals (producers of service) and service recipients (consumers of services). The final principle is to regard service providers as facilitators instead of merely providers to provide service. In the same vein, Redhead et al. [98] proposed several long-term recommendations focusing on equity, ethics, relational health care, and adoption of a life-span approach. Obviously, a systemic approach focusing on the contribution of different systems is needed [99].

## 3. Conclusions

In this review, I start by looking at the negative impact of the pandemic on young people. While there is evidence supporting the negative impact of the pandemic, there are conflicting and inconsistent research findings. I also look at studies on positive mental health in the post-pandemic era. With reference to the related studies in PsycInfo, I highlight several observations, such as comparatively more studies on (a) anxiety and depression than other mental problems; (b) mental health problems than positive mental health; (c) adolescents than emerging adults; (d) Western societies than non-Western societies. I raise several reflections in this paper, including (a) the role of positive psychological attributes; (b) the importance of individual resilience; (c) the importance of family resilience; (d) the need for multidisciplinary, interdisciplinary and transdisciplinary understanding; (e) theoretical refinement directions; (f) future research directions; and (g) intervention directions in the post-pandemic era. Reflection on the pandemic is important. Although the COVID-19 pandemic is not as lethal as the Spanish flu one century ago, the pandemic has had huge repercussions, particularly on the mental health of young people. These reflections will definitely help to prepare young people for the next global pandemic, which will definitely occur, although we do not know exactly when [100].

## Figures and Tables

**Table 1 ijerph-22-01574-t001:** Number of citations on mental health in adolescents or high school students captured by PsycInfo.

Search Terms	Number of Citations in PsycInfo in July 2025)	Number of Citations in PsycInfo (plus “Chinese” as a Search Term in July 2025)
Mental Health Problems		
Depression AND (adolescents OR high school students) AND (post-pandemic OR post-COVID-19)	777	233 ***
Anxiety AND (adolescents OR high school students) AND (post-pandemic OR post-COVID-19)	1877	310 ***
Suicide AND (adolescents OR high school students) AND (post-pandemic OR post-COVID-19)	74	0 ***
Self-harm AND (adolescents OR high school students) AND (post-pandemic OR post-COVID-19)	43	0 ***
Internet addiction AND (adolescents OR high school students) AND (post-pandemic OR post-COVID-19)	46	1 ***
Positive Mental Health		
Resilience AND (adolescents OR high school students) AND (post-pandemic OR post-COVID-19)	76	21 ***
Emotional competence AND (adolescents OR high school students) AND (post-pandemic OR post-COVID-19)	0	0
Hope AND (adolescents OR high school students) AND (post-pandemic OR post-COVID-19)	46	6 ***
Optimism AND (adolescents OR high school students) AND (post-pandemic OR post-COVID-19)	24	6 **
Family resilience AND (adolescents OR high school students) AND (post-pandemic OR post-COVID-19)	7	0 **

Note. One-way Chi-square Test or Goodness-of-Fit Test was performed to examine the differences between the citations with and without “Chinese” as a search term (*** *p* < 0.001; ** *p* < 0.01).

**Table 2 ijerph-22-01574-t002:** Number of citations on mental health in university students or emerging adults captured by PsycInfo.

Search Terms	Number of Citations in PsycInfo in July 2025)	Number of Citations in PsycInfo (plus “Chinese” as a Search Term in July 2025)
Mental Health Problems		
Depression AND (university students OR emerging adulthood) AND (post-pandemic OR post-COVID-19)	470	228 ***
Anxiety AND (university students OR emerging adulthood) AND (post-pandemic OR post-COVID-19)	1725	334 ***
Suicide AND (university students OR emerging adulthood) AND (post-pandemic OR post-COVID-19)	24	0 ***
Self-harm AND (university students OR emerging adulthood) AND (post-pandemic OR post-COVID-19)	23	0 ***
Internet addiction AND (university students OR emerging adulthood) AND (post-pandemic OR post-COVID-19)	48	46
Positive Mental Health		
Resilience AND (university students OR emerging adulthood) AND (post-pandemic OR post-COVID-19)	76	12 ***
Emotional competence AND (university students OR emerging adulthood) AND (post-pandemic OR post-COVID-19)	0	0
Hope AND (university students OR emerging adulthood) AND (post-pandemic OR post-COVID-19)	13	2 **
Optimism AND (university students OR emerging adulthood) AND (post-pandemic OR post-COVID-19)	17	4 **
Family resilience AND (university students OR emerging adulthood) AND (post-pandemic OR post-COVID-19)	10	0**

Note. One-way Chi-square test or Goodness-of-Fit test was performed to examine the differences between the citations with and without “Chinese” as a search term (*** *p* < 0.001; ** *p* < 0.01).

## Data Availability

No new data were created or analyzed in this study. Data sharing is not applicable to this article.

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
