# Peer review of "Mental Health of Young People in the Post-Pandemic Era: Perspective Based on Positive Psychology and Resilience"

_ijerph, 2025, doi:10.3390/ijerph22101574_

Round 1
Reviewer 1 Report
Comments and Suggestions for Authors
Review report
This paper summarizes key insights into youth mental health during adolescence and emerging adulthood, with a focus on the post-pandemic period. The authors acknowledge the significance of inconsistent findings in this field, identify gaps in the literature that require further investigation, and offer suggestions for addressing youth mental health issues.
I have few suggestions that I believe could enhance the quality of the paper:
- The expression „negative mental health“ sounds unusual in scientific discourse. I suggest replacing it with „poor mental health“ or „mental health difficulties/issues“.
- I suggest that the authors consider (among the reasons for the inconsistency of findings) the limited control for pre-pandemic mental health status, which is rarely accounted for in studies, despite evidence of a long-standing increase in mental health problems among youth
- Page 5, lines 141-143: The authors address the distinction between adolescents and emerging adults, but the paper would benefit from a more detailed explanation of the different challenges these two groups face.
- Page 8, line 307: “…of the link between family resilience and perceived pandemic stress.” Is that an error? Did you mean the link between family resilience and mental health?
- Page 8, line 317: This sentence is confusing. I assume the author intended to state that resilience measures are associated with better mental health. I suggest clearly stating this without using double negatives.
- Page 9, line 358 – Did you mean „addictive“ behavior?
- Page 9, line 396 - Define the acronym CAMH in the preceding sentence.
Author Response
Reviewer 1
This paper summarizes key insights into youth mental health during adolescence and emerging adulthood, with a focus on the post-pandemic period. The authors acknowledge the significance of inconsistent findings in this field, identify gaps in the literature that require further investigation, and offer suggestions for addressing youth mental health issues.
I have few suggestions that I believe could enhance the quality of the paper:
Response: I thank the reviewer for his/her assessment.
1. The expression “negative mental health” sounds unusual in scientific discourse. I suggest replacing it with “poor mental health” or “mental health difficulties/issues”.
Response: We have replaced “negative mental health” with “mental health problems” in the revised manuscript.
2. I suggest that the authors consider (among the reasons for the inconsistency of findings) the limited control for pre-pandemic mental health status, which is rarely accounted for in studies, despite evidence of a long-standing increase in mental health problems among youth
Response: We have added this as a possible factor leading to the inconsistency in the related findings in the revised manuscript (page 3, lines 100-101).
3. Page 5, lines 141-143: The authors address the distinction between adolescents and emerging adults, but the paper would benefit from a more detailed explanation of the different challenges these two groups face.
Response: We have added some discussion in the revised manuscript (page 5, lines 151-157).
4. Page 8, line 307: “…of the link between family resilience and perceived pandemic stress.” Is that an error? Did you mean the link between family resilience and mental health?
Response: We have revised this sentence in the revised manuscript (page 8, lines 319-321).
5. Page 8, line 317: This sentence is confusing. I assume the author intended to state that resilience measures are associated with better mental health. I suggest clearly stating this without using double negatives.
Response: We have revised this sentence to make it clearer in the revised manuscript (page 8, lines 330-332).
6. Page 9, line 358 – Did you mean “addictive” behavior?
Response: This is a typo. It has been revised to “addictive” behavior in the revised manuscript (line 372).
7. Page 9, line 396 - Define the acronym CAMH in the preceding sentence.
Response: We have added “child and adolescent mental health” before CAMH (line 411).

Reviewer 2 Report
Comments and Suggestions for Authors
I read Daniel T. L. Shek’s paper entitled “Mental Health of Young People in the Post-Pandemic Era: Observations and Reflections”. Daniel T. L. Shek’s excellent writing skills are evident throughout the text and the reader recognizes this, regardless of whether he agrees or disagrees with the author’s perspective. As this is a Perspective paper, I must admit that the basic characteristics of a Perspective article are successfully fulfilled in this paper. Specifically, the author in an excellent way gives his vision and assessment for the promotion of the mental health of young people in the Post-Pandemic Era, emphasizes future directions, and stimulates thoughts on the possibilities of Positive Psychology and especially resilience. However, the paper falls short in the Analysis of the current research (2.Observations Based on the Related Studies Captured by PsycINFO), where some statistical controls that would justify the author’s arguments are completely absent.
Here are my comments and suggestions:
-Please add the author's perspective to the title; it is necessary to make it clear that the paper is related to positive psychology or resilience.
-Lines 81-83: Please check the sentence, especially the phrase “…enhance their negative mental health”.
-" Observations Based on the Related Studies Captured by PsycINFO": A basic use of statistics would improve the quality of the article.
Author Response
Reviewer 2
Comments and Suggestions for Authors
I read Daniel T. L. Shek’s paper entitled “Mental Health of Young People in the Post-Pandemic Era: Observations and Reflections”. Daniel T. L. Shek’s excellent writing skills are evident throughout the text and the reader recognizes this, regardless of whether he agrees or disagrees with the author’s perspective. As this is a Perspective paper, I must admit that the basic characteristics of a Perspective article are successfully fulfilled in this paper. Specifically, the author in an excellent way gives his vision and assessment for the promotion of the mental health of young people in the Post-Pandemic Era, emphasizes future directions, and stimulates thoughts on the possibilities of Positive Psychology and especially resilience.
However, the paper falls short in the Analysis of the current research (2.Observations Based on the Related Studies Captured by PsycINFO), where some statistical controls that would justify the author’s arguments are completely absent.
Response: I thank the reviewer for his/her positive comment. We have added some statistical analyses in the revised paper.
Here are my comments and suggestions:
1. Please add the author's perspective to the title; it is necessary to make it clear that the paper is related to positive psychology or resilience.
Response: The title has been revised.
2. Lines 81-83: Please check the sentence, especially the phrase “…enhance their negative mental health”.
Response: We have revised the sentence to make it clearer (page 2, lines 80-82).
3. " Observations Based on the Related Studies Captured by PsycINFO": A basic use of statistics would improve the quality of the article.
Response: We have added some statistical analyses to reinforce the observations (pages 3 and 4, tables 1 and 2).
